# Studies on the Drug Loading and Release Profiles of Degradable Chitosan-Based Multilayer Films for Anticancer Treatment

**DOI:** 10.3390/cancers12030593

**Published:** 2020-03-05

**Authors:** Hyeongdeok Sun, Daheui Choi, Jiwoong Heo, Se Yong Jung, Jinkee Hong

**Affiliations:** 1Department of Chemical and Biomolecular Engineering, Yonsei University, Seoul 03722, Korea; sunbbang1332@yonsei.ac.kr (H.S.); daheui.choi@yonsei.ac.kr (D.C.); jw.heo88@yonsei.ac.kr (J.H.); 2Division of Pediatric Cardiology, Department of Pediatrics, Yonsei University College of Medicine, Seoul 03722, Korea

**Keywords:** multilayer nanofilm, polysaccharide, tannic acid, layer-by-layer, drug loading and release behavior, anticancer treatment

## Abstract

This study demonstrates the possibility of developing a rapidly degradable chitosan-based multilayer film for controlled drug release. The chitosan (CHI)-based multilayer nanofilms were prepared with three different types of anions, hyaluronic acid (HA), alginic acid (ALG) and tannic acid (TA). Taking advantage of the Layer-by-Layer (LBL) assembly, each multilayer film has different morphology, porosity and thickness depending on their ionic density, molecular structure and the polymer functionality of the building blocks. We loaded drug models such as doxorubicin hydrochloride (DOX), fluorescein isothiocyanate (FITC) and ovalbumin (Ova) into multilayer films and analyzed the drug loading and release profiles in phosphate-buffered saline (PBS) buffer with the same osmolarity and temperature as the human body. Despite the rapid degradation of the multilayer film in a high pH and salt solution, the drug release profile can be controlled by increasing the functional group density, which results in interaction with the drug. In particular, the abundant carboxylate groups in the CHI/HA film increased the loading amount of DOX and decreased rapid drug release. The TA interaction with DOX via electrostatic interaction, hydrogen bonding and hydrophobic interaction showed a sustained drug release profile. These results serve as principles for fabricating a tailored multilayer film for drug delivery application.

## 1. Introduction

Development of smart drug delivery system (DDS) for the controlled drug loading and release manner has been considered essential to solving drug dosage, efficacy and safety issues. Drug concentrations above the toxic threshold (maximum safe level) would cause harm to the patient, whereas those below the subtherapeutic threshold (minimum effective level) are ineffective [1,2]. Therefore, drug release should be predictable for application between therapeutic levels [3]. Controllable drug loading and release is advantageous because it reduces the drug dosage, minimizes side effects and toxicity, extends the therapeutic activity time, protects sensitive drugs and increases the therapeutic effect [4,5,6]. However, drug delivery is not easily controlled; therefore, an effective platform is needed for containing and providing controlled release of the drug. Among the various platforms preparation technique for controlling drug release, the layer-by-layer (LBL) assembly is suitable for smart drug delivery [7,8,9,10,11]. This technique is subsequent deposition of oppositely charged polyelectrolytes in a molecular level [12]. Various polyelectrolytes can be combined to fabricate multilayer films whose pore size, porosity, thickness and structure can be easily controlled by adjusting the conditions of the polymer solution such as pH and salt concentration [13,14,15,16,17,18].

A drug delivery platform also requires a highly biocompatible material [19]. Natural polymers are commonly used over synthetic polymers for drug delivery applications because of their proven biocompatibility and biodegradation [6,20,21]. However, the degradable properties of natural polymers and the high pH and salt level in vivo lead to a significant decrease in the stability of a multilayer film, resulting in an inadvertent drug release from the film. In previous studies, because of the disadvantages of multilayer films made of natural polymers, multilayer films were crosslinked or used hydrolytically degradable synthetic polymers [22,23].

In the present study, we studied the film structures, stability and functional group densities of chitosan-based multilayer films and their effect on drug loading and release profiles in environments with the same osmolarity and temperature as the human body. To fabricate a drug delivery platform with biocompatible and controlled drug release, we designed chitosan (CHI)-based multilayer films in which the CHI is combined separately with hyaluronic acid (HA), alginic acid (ALG) and tannic acid (TA). CHI is an N-deacetylation product of chitin, which is a natural polysaccharide, composed of glucosamine and N-acetyl glucosamine units [24], and is the positively charged polyelectrolyte in polysaccharides. HA contains repeating units of disaccharide, β-1,4-d-glucuronic acid-β-1,3-N-acetyl-d-glucosamine and is a natural polysaccharide found in the tissues of living organisms [24]. In addition, HA has a carboxyl group per repeating unit of disaccharide. ALG is a natural polysaccharide extracted from marine algae, and is composed of one to four linked α-l-guluronic and β-d-mannuronic acid [24]. The carboxyl groups on repeating units provide the negative charges in ALG. Although TA is not a polysaccharide, it is a natural polyphenol plant-derived substance containing cyclic glucose and digalloyl groups. In addition, TA is effective in cancer therapy and has antibacterial properties [25,26].

In this work, we design a CHI/HA multilayer film with a nanoporous structure through which small molecule drugs can diffuse. CHI diffuses through the interlayer of the multilayer film, which induces vacancies [27,28]. In the case of the CHI/ALG multilayer film, it has a dense structure and abundant carboxyl groups. The carboxyl group present in each monosaccharide repeating unit per ALG provides more abundant carboxyl groups than other combinations. The CHI/TA multilayer film is designed to have a macroporous structure. The entangled CHI binds to the TA by hydrogen interaction, forming a rough and macroporous structure. Moreover, the small-molecule model, macromolecule model and anticancer drugs such as fluorescein isothiocyanate (FITC), ovalbumin (Ova) and doxorubicin hydrochloride (DOX) are loaded onto each film and the drug release profile is analyzed. The various drug release profiles and properties of each multilayer film are combined to study the parameters affecting the drug release profile. As a result, all combinations released the drug rapidly within 30 min, with the film degradation.

## 2. Results and Discussion

### 2.1. Characterization of Multilayer Films

In this study, we fabricated CHI-based CHI/HA, CHI/ALG, and CHI/TA films on separate Si wafers (Figure 1B). The multilayer film thickness growth curves and AFM images for each multilayer film are shown in Figure 2 and Figure 3. All multilayer films initially grew linearly before four bilayers and then changed exponentially with the number of layers (Figure 2, Appendix A). This is because the first few layers are deposited close to the Si wafer and affected by the Si wafer, resulting in slightly thinner than subsequent layers [14,29]. In the CHI/HA and CHI/ALG films, CHI chains acted as the diffusing species through the multilayer films, leading to improved accessibility of small molecules and exponential growth [27,28,30]. First, for the CHI/HA film, we controlled the salt concentration in the polyelectrolyte solution by electrostatic interaction to fabricate vacancies such as those in the nanoporous structure of the film. The salt screens the ionization sites on polyelectrolytes and reduces the repulsion of intra-polyelectrolytes [31]. CHI chains with reduced repulsive force build a film by entangling and diffusing the CHI chain into a multilayer film, resulting in a nanoporous structure and exponential growth of the CHI/HA film. Second, to fabricate the dense structure of the CHI/ALG film by electrostatic interaction, we controlled the pH of the polyelectrolyte solution. The ionization of the polymer chain can be controlled by pH. Therefore, we adjusted the pH of CHI solution below the pKa value of pH 6.5. because an almost ionized CHI chain can build a dense film by extending the CHI chain and inhibiting the diffusion through the multilayer film [28,32]. In addition, we controlled the pH of the ALG solution to 3.5, which is similar to the pKa value (ranges from 3.4 to 3.7) because of the relatively thicker multilayer film fabricated as a drug reservoir [33]. Third, to fabricate the macroporous structure of the CHI/TA film by hydrogen bonds, we controlled the pH of the CHI solution above that of the pKa value (pH 6.5) because the rarely ionized CHI can bind with hydroxy groups of TA via hydrogen bonds. The CHI chain with rare ionization at pH 7 (about 30%) builds a film through highly entangled chains, resulting in a highly rough structure and exponential growth in the CHI/TA film [33,34].

The buildup of the three combinations of multilayer films was analyzed by AFM as the number of layers increased. In the early steps of the deposition, the CHI/HA film exhibited small islets with sizes from several hundred nanometers to 1 micrometer [28]. As the number of layers increased, the small islet surface changed to a smooth surface (Figure 3A). The change in surface resulted from the diffusion of CHI chains through the multilayer film. The CHI inside the multilayer film moved to the surface and formed a new bond with the HA, which changed the film’s surface to smooth [28]. The CHI/ALG film also had islets, which increased in size and roughness as the number of layers increased (Figure 3B). This occurred because a highly ionized CHI chain leads to low diffusion through the film and dense layer deposition. Although the thickness of the CHI/ALG film grew exponentially, it had a relatively dense structure because it did not have vacancies formed by the diffusion of CHI chains and was thinner than other films with the same number of layers. The surface morphology of the CHI/TA film is shown in Figure 3C. The roughness values (Rq) of the CHI/HA, CHI/ALG, and CHI/TA films were 60.16 nm, 158.54 nm and 192.84 nm, respectively. High surface roughness was observed in the CHI/TA film because the rarely ionized CHI chain enabled the CHI/TA multilayer to form a rough and macroporous structure (Appendix A). The macroscopic images of CHI/TA film revealed that macro-size clumps are formed, whereas non- or barely coated areas also existed.

### 2.2. Film Degradability

The degraded thickness of the multilayer film on the Si wafer in 1× PBS (pH 7.4, 1.5 mM of KH_2_PO_4_, 0.15 M of NaCl, 2.7 mM of Na_2_HPO_4_-7H_2_O) at 37 °C was measured using the profilometer (Figure 4A). The surface morphology and roughness of the degraded multilayer films were also characterized by AFM (Figure 4B,C). We have set the thickness of multilayer films from 500 nm to 1 μm to facilitate a comparison of film degradability. Before measuring, we have prepared each film has same thickness, i.e., five, 12 and 10 bilayers for HA, ALG and TA films, respectively, were prepared. As shown in Figure 4A and Appendix A, the (CHI/HA)_5_ and (CHI/ALG)_12_ films degraded rapidly within 30 min, and the thickness decreased by 67% and 63%, respectively. because the pH and salt concentration of PBS affected the stability of the multilayer film stability. At pH 7.4, HA and ALG chains are highly ionized, at almost 100%. The highly ionized polymer chains created an electrostatic repulsion within the film layers, resulting in decreased multilayer film stability [35]. The capacity of the CHI to complex with ions can also affect the multilayer film stability [36,37]; the multilayer film was not completely erased because the salt in the PBS reduces the ionic strength by screening the ionization site [38]. The (CHI/TA)_10_ film degraded gradually, and the thickness decreased by 55% within 30 min (Figure 4A and Appendix A). Because the (CHI/TA)_10_ film was built by clumps of TA linked to CHI chains (CHI+TA aggregates), some TA molecules inside the clumps can be released [39,40]. The surface morphologies of the (CHI/HA)_5_ film and (CHI/ALG)_12_ film after degradation exhibited new islet shapes. The dislinking and disassembly of CHI+TA aggregates, which are indicated in Appendix A (white arrows), clearly existed, demonstrating that the CHI/TA films are degraded in the macroscopic range. Considering the AFM images in Figure 4B and the roughness of the degraded multilayer film in Figure 4C, we can confirm rapid decreases in the film thickness and the roughness of film surface after treatment in 1× PBS at 37 °C.

### 2.3. Film Characterization of Functional Group Followed by Fourier-Transform Infrared Spectroscopy

The film characteristics for functional groups were investigated by Fourier-Transform Infrared Spectroscopy (FT-IR) according to the spectra after deposition of each layer in the (CHI/HA)_5_, (CHI/ALG)_12_ and (CHI/TA)_10_ films (Figure 5). Three main peaks were identified. The characteristic saccharide peaks in the range of 900–1200 cm^−1^ represent the polysaccharide backbone vibrations and contain C‒O stretching at 1082 cm^−1^, 1032 cm^−1^ and 1159 cm^−1^ [41,42]. The area between 1400 and 1500 cm^−1^ contains carboxylate peaks for HA and CHI, and the peak at 1412 cm^−1^ is the -COO^−^ symmetric stretching from HA and ALG [23,43]. The area around 1600 cm^−1^ contains primary amide peaks for HA and CHI. The peaks at 1601 cm^−1^, 1620 cm^−1^ and 1595 cm^−1^ are attributed to -COO^−^ asymmetric stretching of HA, C=O vibration of CHI and N‒H bending of CHI [23,44]. The characteristic TA peaks appeared at about 1450 cm^−1^ for the C‒C aromatic group and 1718 cm^−1^ for the carbonyl groups [45].

We investigated the functional group density according to the IR spectra because the height of the IR peak indicates the intensity of the molecular bonds in the multilayer films. We compared the carboxylate and amine peaks of each multilayer film, which indicate the amount of ionization within the films and are directly related to the charge of the drug loading and release capacity. The carboxyl groups in HA and ALG can be easily ionized and are negatively charged, which can affect the incorporation or release of positively charged drugs. The (CHI/ALG)_12_ film had the highest carboxylate group peak at 1412 cm^−1^. This means that more carboxylate groups are contained in multilayer films. We believed that the increased carboxylic acid groups in films drove the positively charged drug incorporation within the multilayer films. The primary amine in CHI can be easily ionized and is positively charged, which can affect the incorporation or release of positively charge drugs. The (CHI/ALG)_12_ film had the highest primary amine peak, at 1595 cm^−1^.

### 2.4. Characterization of Drug Loading and Release from Multilayer Films

To measure the drug release from the three different types of multilayer films, FITC, Ova, and DOX were loaded separately into the respective films. Figure 6 and Figure 7A show the drug release profiles in 1× PBS at 37 °C. Rapid drug release was observed for all multilayer films. As shown in the normalized drug release graphs (Appendix A), the half-release time points for FITC, Ova and DOX are 0.35 h, 0.32 h and 0.29 h for the CHI/HA film, 0.34 h, 0.42 h and 0.51 h for the CHI/ALG film, and 0.45 h, 0.34 h and 2.43 h for the CHI/TA film, respectively. Rapid drug release was observed for all multilayer films, with almost complete degradation occurring within 30 min (Figure 4A); most of the drugs were released during the same period. These results indicate that the rapid drug release was most likely cause by the rapid degradation of the multilayer films. Figure 6A shows the release profile of FITC, which is a negatively charged, small-molecule drug model with a pH of 7.4. Thus, FITC represents the loading and release behavior of small molecular drugs [46]. The rapid release profile of FITC- loaded CHI/HA and CHI/ALG films shown in Appendix A indicates that the FITC was loaded with no binding to the polymer chain and was released by diffusion. The reason for the abundant amount of FITC loaded in the CHI/HA film is the high swelling ratio in the PBS environment [47,48]. The least amount of FITC was loaded into the CHI/ALG film, owing to the drug’s repulsion toward the many carboxylate groups of ALG (Figure 5). The CHI/TA film inhibited ionization by binding the TA and primary amine groups of CHI through the multilayer deposition process [39]. FITC can have electrostatic interactions with a few ionized primary amines of CHI in CHI/TA film. Therefore, the FITC-loaded CHI/TA film release profile shows a relatively smaller amount than that in the other multilayer films during the initial 30 min (Appendix A). This indicates that the CHI/TA film bound with the FITC, which was not affected by the degradation of the multilayer film and diffusion in the PBS.

Figure 6B shows the release profile of Ova, which was used as a drug model to represent negatively charged macromolecules under physiological conditions [49]. The CHI/HA and CHI/TA films were loaded with abundant Ova loading property, 2.8 and 2.5 times higher than that of the CHI/ALG film. This is because these film structures include pores, which facilitated diffusion of the macromolecule Ova through the multilayer films. The Ova release profile also shows rapid diffusion (Appendix A). The CHI/ALG film had the least loaded Ova owing to the drug’s repulsion toward the many carboxylate groups in the multilayer film (Figure 5). The dense structure of the CHI/ALG film inhibited the diffusion of the Ova, resulting in a relatively sustained release after the initial 30 min (Appendix A).

Figure 7A shows the release profile of DOX, which is an anticancer drug with positively charged small molecules with a pH of 7.4. The small molecules have high mobility in a buffered environment, and the multilayer film structure does not significantly affect the release profile [10,50]. The DOX incorporation into the multilayer film showed large differences in the release profile depending on the density and charge of the functional group. The CHI/ALG film had the largest amount of DOX, which corresponded to its high carboxylate group density according to the FTIR results (Figure 5). The amount of DOX released in the CHI/ALG film was relatively smaller than that in the other drug release profiles during the initial 30 min (Appendix A) owing to the DOX binding of this drug with the carboxylate group of the CHI. The CHI/HA film also has a carboxylate group of HA; however, the amount of DOX loaded into the CHI/HA film was lower than that for the CHI/ALG film because the carboxylate group density was low. Rapid release of the DOX loaded in the CHI/HA film was observed in the release profile (Appendix A); however, that loaded onto the CHI/TA film was a small amount, and a sustained release profile was shown. For the CHI/TA film, the main interaction between TA and DOX is a hydrophobic interaction, so we hypothesized that the CHI/TA film would induce higher DOX incorporation efficiency as well as trapping efficiency [51]. However, the highly rough structure of the CHI/TA film (Appendix A) that is generated by bulk CHI+TA aggregates resulted in a lower chance of binding with DOX. The merged confocal image of the DOX in CHI/TA film demonstrated that the DOX is only located in aggregates (Figure 7B). Also, sustained DOX release indirectly revealed that the DOX still binds with TA via a hydrophobic interaction in PBS conditions (pH 7.4), which is different pH and ionic strength from the film deposition conditions (Appendix A). At pH 7.4, TA has a negative charge and can have a strong interaction with DOX via an electrostatic interaction, hydrophobic interaction or hydrogen bonding [51]. The surface morphology of the DOX-loaded multilayer films was measured using a confocal laser scanning microscope. The DOX loading was nonhomogeneous in all of the multilayer films (Figure 7B). Strong fluorescence intensity was observed in the CHI/ALG film. This indicates a larger amount of DOX loading than for the other multilayer films, corresponding to the DOX release profile (Figure 7A). Considering the DOX release profile in Figure 7A and the surface morphology images of the DOX-loaded multilayer film in Figure 7B, we confirm that the most important factor for the incorporation and release of drugs into CHI-based multilayer films is the interaction between the drug and the functional group density.

### 2.5. Film Toxicity and Anticancer Effect Analysis

To verify that the drug release from the multilayer films maintained its functionality, we investigated the anticancer effects of DOX-loaded multilayer films for cancer cells (Figure 8). In this study, we used HeLa and KATO III cells, which are cervical cancer cells and gastric cancer stem cells, respectively. In all cases, the DOX-loaded multilayer films were effective anticancer systems, even though the bare CHI/TA film showed an anticancer effect. Also, for HeLa cells, the anticancer effect was proportional to the amount of DOX incorporation, resulting in decreased viability of a 42.60%, 96.63% and 41.57% for the CHI/HA, CHI/ALG, and CHI/TA films, respectively. Compared to HeLa cells, KATO III cells showed resistance to the DOX. The CHI/ALG film contained more DOX than other combinations, resulting in the highest anticancer effect for HeLa cells. In particular, the CHI/TA film had an anticancer effect regardless of DOX incorporation because TA has anticancer effects through inhibiting cancer cell proteasome activity and inducing G1 arrest and apoptosis [26]. Therefore, the CHI/TA film was loaded with the least amount of DOX, but the synergy effect with TA and degradable properties of CHI/TA film resulted in 28% HeLa cell viability. In terms of the viability of KATO III cells (Figure 8B), the anticancer effects of DOX in CHI/ALG and CHI/TA films in KATO III cell were slightly different to those in HeLa cells. The CHI/ALG film contained more DOX than other films, but anticancer effect was the highest in the CHI/TA film. KATO III cells have DOX resistance, whereas TA directly inhibits the formation of cancer stem cells [52,53]. The CHI/HA showed no anticancer effect in either HeLa or KATO III cells due to the small amount of DOX. In terms of the safety, the CHI/HA film and CHI/ALG were nontoxic to normal human dermal fibroblast (HDF) cells, but the CHI/TA film showed toxicity to HDF cells (Appendix A). The dose of DOX and degradable properties of CHI/TA should be optimized to reduce the toxicity and maintain the anticancer effect. In the drug release profile, a DOX-incorporated CHI/TA film released DOX 0.15 μg for 24 h, and showed lower HeLa and KATO III cell viability compared to the control with DOX 0.5 μg. This means that CHI/TA films being used as a DOX delivery platform would reduce the DOX dose.

## 3. Materials and Methods

### 3.1. Materials

CHI (M_w_ = 190,000‒310,000 g/mol), ALG (M_w_ = 120,000‒190,000 g/mol), TA, DOX, and FITC were purchased from Sigma-Aldrich (St. Louis, MO, USA). Mg^+^ and Ca^2+^-free 10× PBS was purchased from Gibco (Waltham, MA, USA). The hyaluronic acid was purchased from SK Bioland (Cheonan, Chungnam, South Korea). The Ova was purchased from Bio Basic Inc. (Konrad Crescent, Markham, Ontario, Canada). The Ova fluorescein conjugate was purchased from Invitrogen (Carlsbad, CA, USA).

### 3.2. Preparation of Multilayer Films

We fabricated the three different types of multilayer films using CHI as a positively charged polymer and HA, ALG, and TA as negatively charged polymers. CHI was used to prepare separate multilayer nanofilms with HA (CHI/HA), ALG (CHI/ALG) and TA (CHI/TA). Each multilayer film was fabricated on an Si wafer using the LBL assembly technique [12,32,54]. The Si wafer was treated with O_2_ plasma for 2 min to modify the negatively charged surface. The substrate was first immersed into the CHI solution for 5 min and then washed for 1 min twice with deionized water. The substrates were then moved into the respective HA, ALG and TA solutions, deposited for 5 min, and washed with deionized water. This process was repeated until the desired number of layers was deposited. After deposition, the films were dried with N_2_ gas and stored at room temperature. The concentration of each polymer solution was 2 mg/mL in deionized water, as shown in Figure 1. For fabricating the CHI/HA film, the HA and CHI were dissolved in 0.05 M NaCl in water. For fabricating the CHI/ALG film, the pH was 3.5 and 6 for the CHI and ALG solutions, respectively; that for the CHI solution in the CHI/TA film was 7.

### 3.3. Characterization of Multilayer Films

The thickness of the multilayer film growth was measured using a profilometer (Dektak 150, Veeco, Plainview, NY, USA). the film morphology and roughness were measured using atomic force microscopy (AFM; NX10, Park Systems, South Korea). The film characteristics for functional groups were investigated by Fourier-Transform Infrared Spectroscopy (IR-4700, Jasco, Hachioji, Tokyo, Japan) using dried CHI-based films. The infrared spectra of film characteristics of functional groups was recorded on a Jasco spectrometer by averaging 32 scans with a resolution of 4 cm^−1^. For quantitative analysis of the deposition of each layer on the Si wafer, we used a quartz crystal microbalance (QCM; QCM200, Stanford Research Systems, Sunnyvale, CA, USA). For confirmation of the film stability, degradation of the multilayer films was conducted under physiological conditions. In this process, each multilayer film was immersed in 1× PBS and then placed in an incubator at 37 °C to provide the same osmotic pressure and temperature as the human body. At different time points, the films were washed with deionized water to remove salts and dried. The film thickness was then measured using a profilometer.

### 3.4. Characterization of Drug Loading and Release from Multilayer Films

In this study, we used FITC and Ova as small-molecule and macromolecule model drugs, respectively. In addition, we used DOX as a drug for cancer treatment. To detect the fluorescence of Ova, we mixed fluorescent-labeled Ova with nonfluorescent Ova at a ratio of 1:9. To incorporate the drug into the multilayer films, 0.5 mg/mL of each drug solution was added to the multilayer films. After 1 h of incorporation, the multilayer film was washed with deionized water to remove the additional drugs.

Afterward, the drug-incorporated multilayer CHI/HA, CHI/ALG and CHI/TA films were immersed into 1 mL of 1× PBS and were then placed in an incubator at 37 °C. At determined points in time, 1 mL of each drug containing PBS was collected, and 1 mL of fresh PBS was added. The amount of drug released from each multilayer film was measured using a fluorescence plate reader (Varioskan Flash 3001, Thermo Fisher Scientific, Vantaa, Finland). The normalized drug release profiles were calculated by division the amount of drug released at different determined time points by amount of drug release for 34 h. To confirm the incorporation of the drug into the multilayer films, we used a confocal laser scanning microscope (LSM 880; Carl Zeiss, Oberkochen, Germany). The same laser conditions were used to irradiate all samples.

### 3.5. Cell Culture

HDF (human dermal fibroblasts), HeLa and KATO III cells were cultured in a 100-mm culture dish (SPL Life Sciences Co., Ltd., Pocheon-si, South Korea) and incubated at 5% CO_2_ incubator. After two to three days of cultivation, the cells were detached by trypsinization using 0.05% trypsin-EDTA in PBS. The cells were harvested by centrifugation at 1500 rpm for 3 min. The complete HeLa culture medium was composed of 90% high glucose Dulbecco’s modified eagle medium (DMEM), 10% fatal bovine serum (FBS) and 1% penicillin‒streptomycin (PS) for HeLa and HDF medium. For KATO III cells, we used RPMI 1640 medium instead of DMEM. All reagents were purchased from Gibco except FBS (Welgene, Gyeongsan-si, South Korea).

### 3.6. Film Toxicity and Anticancer Effect Analysis

HDF, HeLa and KATO III cells were seeded onto 24-well plates at a density of 1 × 10^4^ cells/well. After one day of cultivation, the multilayer films prepared on the Si wafers were soaked in culture media. After 24 h treatment, we removed the films and 10% cell counting kit-8 (CCK-8; Dojindo, Kumamoto, Japan) was added to the cells. The absorbance of changed color by cell metabolism was measured at 450 nm visible wavelength using a SpectraMax 340 PC plate reader (Molecular Devices, San Jose, CA, USA) after 2 h incubation at 37 °C. The cells were treated with 20% dimethylsulfoxide (DMSO) as a positive control.

### 3.7. Statistical Analysis

We analyzed the differences between two sets of data using a two-tailed *t*-test for control and experimental data. The differences were statistically significant at the 0.1%, 1% or 5% level.

## 4. Conclusions

In this study, we fabricated three different types of CHI-based multilayer films. Each had different characteristics depending on the building mechanism and material constitution. The CHI/HA film had a nanoporous structure with diffusion of the CHI chain during deposition; the CHI/ALG film had a dense structure with the deposition of highly ionized ALG chains and many carboxylate groups in multilayer film; and the CHI/TA film had a macroporous structure with the deposition of rarely ionized CHI chains. Various aspects of film characterization were measured such as thickness, surface morphology, degradation, drug loading and release. We characterized different drug release profiles based on the film properties. The film degradation was the main parameter for drug release. Degradation and drug release were rapid during the initial 30 min in the PBS, which indicates that film degradation affected the drug release. However, multiple functional groups of the multilayer film led to an interaction between the drug and the film, which inhibited the rapid drug release for the initial 30 min, even when multilayer film degradation occurred. Therefore, favorable interactions between drug and film would sustain the diffusion of drugs through the multilayer film. Finally, the anticancer drug-incorporated CHI-based film showed various anticancer properties depending on the amount of DOX incorporation and film properties. In this study, we found that the CHI-based multilayer films are a good drug delivery system for DOX, controlling the drug loading and release kinetics easily depending on the properties of building blocks, and thus fine-tuned therapeutic efficacy is feasible. Therefore, we anticipate that a polysaccharide multilayer film containing many functional groups and capable of binding to small-molecule drugs could control drug loading and release under physiological conditions. The CHI-based multilayer films showed potential as a cancer treatment platform due to their precise control of drug loading and release kinetics.

## Figures and Tables

**Figure 1 cancers-12-00593-f001:**
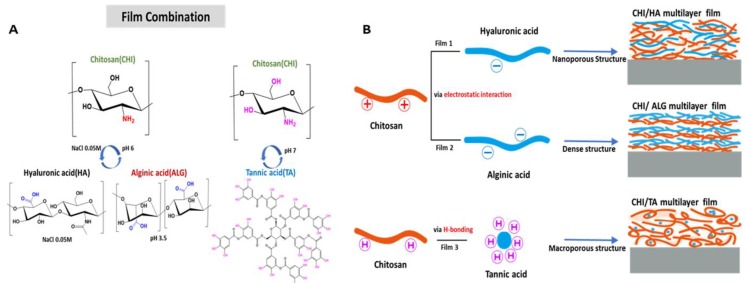
(**A**) Chemical structure of materials showing the combination and deposition conditions of each multilayer film. (**B**) Schematic illustration of preparation of each multilayer film. The CHI/HA and CHI/ALG films were fabricated via electrostatic interaction, whereas the CHI/TA film was fabricated via hydrogen bond and electrostatic interaction. Each film has a different internal structure depending on the driving forces and ionic density.

**Figure 2 cancers-12-00593-f002:**
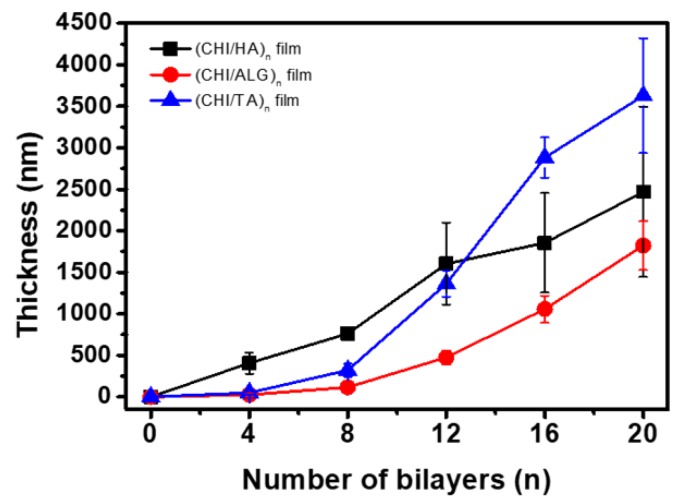
Growth curve of the multilayer film thickness on the Si wafer as the number of layers increased.

**Figure 3 cancers-12-00593-f003:**
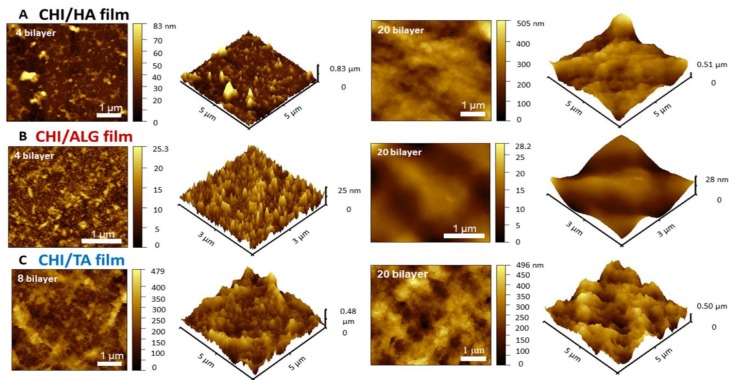
Surface morphology changes in (**A**) CHI/HA, (**B**) CHI/ALG, and (**C**) CHI/TA films as the number of layers increased, as measured by AFM.

**Figure 4 cancers-12-00593-f004:**
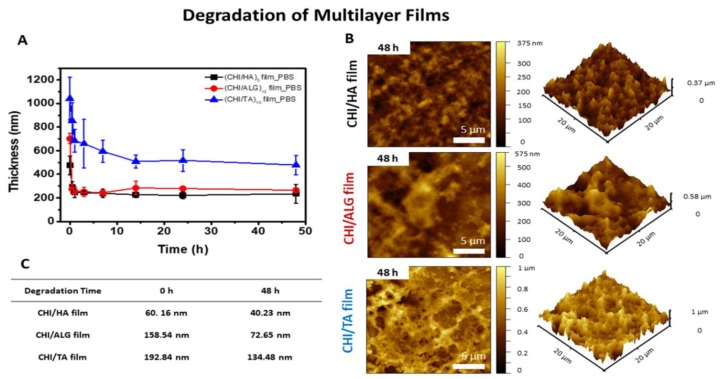
(**A**) Degradation rate of multilayer films under 1× PBS at 37 °C measured by changes in thickness. (**B**) Surface morphology of multilayer films under 1× PBS at 37 °C for 48 h. (**C**) Surface roughness of multilayer films measured by AFM.

**Figure 5 cancers-12-00593-f005:**
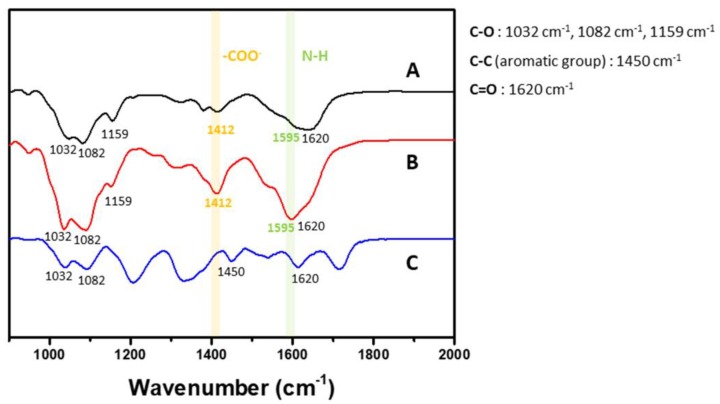
FTIR spectra of (**A**) (CHI/HA)_5_, (**B**) (CHI/ALG)_12_, and (**C**) (CHI/TA)_10_ films.

**Figure 6 cancers-12-00593-f006:**
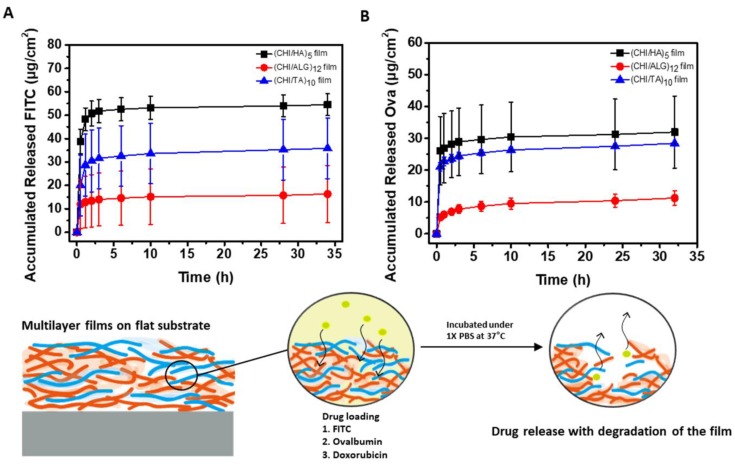
Accumulated amount of released (**A**) FITC and (**B**) Ova from multilayer films under 1× PBS at 37 °C.

**Figure 7 cancers-12-00593-f007:**
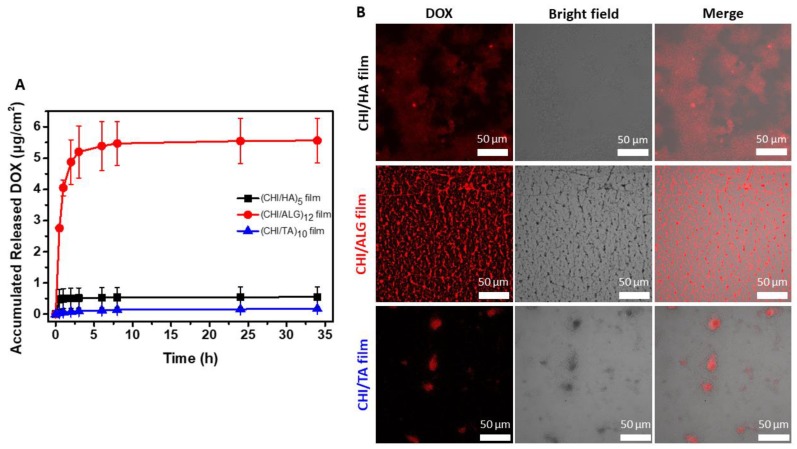
(**A**) Accumulated amount of released DOX from multilayer film under 1× PBS at 37 °C. (**B**) Confocal microscope image of DOX-incorporated multilayer films. Red fluorescence and bright field represent DOX and film structure. To clearly see the DOX fluorescence in the film, we have adjusted the red fluorescence balance. The original images are displayed in Appendix A.

**Figure 8 cancers-12-00593-f008:**
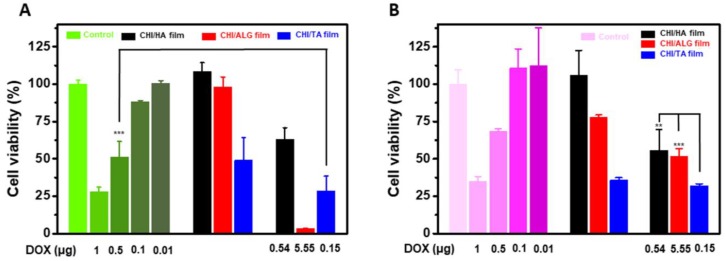
Film toxicity and anticancer effect of DOX-incorporated multilayer films. (**A**) Cell viability of HeLa cells and (**B**) KATO III cells. ** and *** suggest that *p* value is smaller than 0.01 and 0.001, respectively.

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
