# Peer review of "Studies on the Drug Loading and Release Profiles of Degradable Chitosan-Based Multilayer Films for Anticancer Treatment"

_cancers, 2020, doi:10.3390/cancers12030593_

Round 1

Reviewer 1 Report

In this study, the authors have demonstrated the loading and release mechanism of 3 different kinds of model drugs using layer-by-layer (LbL) assembled multilayer nanofilms. The chitosan, which is biocompatible and nature-derived polymer, is mainly used for ingredient of nanofilms. 3 different kinds of chitosan based LbL films were prepared, and those have different growth curves, morphology, and chemical groups because of the ionic density, molecular weight and structure of each building block. Finally, those kinds of different properties resulted in different drug loading and release kinetics. Specially the alginate multilayer film showed larger amount of doxorubicin incorporation and release capacity. The controlled drug loading and release using LbL film looks novel, however some lack of points should be revised.

  1. In line 95, the multilayer films grow linearly at the first 4 bilayers and changed to exponential growth. Why it is that? The author should explain the phenomena.
  2. In line 191, the half life of drug release for each film looks quite different in figure S3. Revise it.
  3. The amount of DOX in each film is extremely different. The ALG film have DOX 5-6 times larger than other films due to their carboxylic acid groups in film. However, DOX has lot of benzene rings than amine group. It means that it is more chance to bind with TA by hydrophobic interaction. Why this phenomenon was not happened?

Author Response

Response to Reviewer 1 Comments

In this study, the authors have demonstrated the loading and release mechanism of 3 different kinds of model drugs using layer-by-layer (LbL) assembled multilayer nanofilms. The chitosan, which is biocompatible and nature-derived polymer, is mainly used for ingredient of nanofilms. 3 different kinds of chitosan based LbL films were prepared, and those have different growth curves, morphology, and chemical groups because of the ionic density, molecular weight and structure of each building block. Finally, those kinds of different properties resulted in different drug loading and release kinetics. Specially the alginate multilayer film showed larger amount of doxorubicin incorporation and release capacity. The controlled drug loading and release using LbL film looks novel, however some lack of points should be revised.

We would like to thank the reviewer for thorough reading of the manuscript and helpful comments. We have revised our manuscript according to your helpful comments. The following is point-by-point responses to the reviewer’s comments.

Point 1: In line 95, the multilayer films grow linearly at the first 4 bilayers and changed to exponential growth. Why it is that? The author should explain the phenomena.

Response 1: Thank you for the thoughtful suggestion. We already explained that why each combination has exponential growth manner in lines 96 and 113. Therefore, in consideration of the general principle for the multilayer film assembly, we could explain that why the film grows linearly and then grows exponentially.

The principle of the multilayer film assembly is that each polyelectrolyte layer continuously forms a multilayer film. The properties of the multilayer film are presented by the polyelectrolyte layer that accumulates layer-by-layer on the substrate, and the properties of the multilayer film can be affected by various factors such as substrate and solution condition. Previous researchers have found that there are precursor layers (for the first several layers), which behave differently than subsequent layers (Langmuir (2000) 16, 1249-1255, Langmuir (2001) 17, 6655-6663). There would be differences between the first few layers and subsequent layers. The first layer is deposited close to the substrate which are largely influenced by the substrate, and they will be formed a substantially different structure. Generally, the thickness of precursor layers is slightly thinner than subsequent layers. However, where the precursor layer end is still not clear. In our study, we found that all multilayer films were fabricated grows exponentially after linear growth up to 4 bilayers affected by the substrate.

For the better understanding to readers, we revised regarding the film growth mechanism on manuscript.

[Page 3, Section 2.1, line 95]

All multilayer films initially grew linearly before 4 bilayers and then changed exponentially with the number of layers (Figure 2, S1† and S2†). Because the first few layers are deposited close to the Si wafer and influenced by the Si wafer, resulting in slightly thinner than subsequent layers (Langmuir (2000) 16, 1249-1255, Langmuir (2001) 17, 6655-6663).

Point 2: In line 191, the half life of drug release for each film looks quite different in figure S3. Revise it.

Response 2: Thank you for the comment. We agreed that the half-life of drug release profile for each multilayer film quite different. We revised it based on figure 3S.

[Page 6, Section 2.4, line 197]

Rapid drug release was observed for all multilayer films. As shown in normalized drug release graphs, figure S3, the half-release time points for FITC, Ova and DOX are 0.35 h, 0.32 h, and 0.29 h for CHI/HA film, 0.34 h, 0.42 h, and 0.51 h for CHI/ALG film, and 0.45 h, 0.34 h, and 2.43 h for CHI/TA film, respectively.

Point 3: The amount of DOX in each film is extremely different. The ALG film have DOX 5-6 times larger than other films due to their carboxylic acid groups in film. However, DOX has lot of benzene rings than amine group. It means that it is more chance to bind with TA by hydrophobic interaction. Why this phenomenon was not happened?

Response 3: Thank you for the thoughtful suggestion. We also agreed with reviewer’s comment. Indeed, the previous our publication regarding on the DOX-incorporated microcapsules composed of gelatin and TA suggested that strong hydrophobic interaction between gelatin, TA and DOX resulted in higher DOX incorporation efficiency as well as trapping efficiency (Nanoscale, (2018) 10, 18228-18237.). Therefore, according to the previous study, we firstly hypothesized that the CHI/TA film has a highest DOX-incorporation capacity via hydrophobic interaction. However, the DOX release result represented the CHI/TA film dose not have much DOX-incorporation capacity.

The main reason of this phenomena is due to the film growth behaviour of CHI/TA film. Different from other film, CHI/TA film grew faster with large standard deviation, which means that the film is highly rough. Also, in AFM and SEM image (figure 3 and S3), the CHI/TA multilayer film has rough structure with bulk CHI+TA aggregates. The non- or little coated film zones are also exhibited. Therefore, we found that the uneven CHI/TA multilayer film were prepared due to the strong CHI+TA aggregate’s creation, resulting that the lower moieties to bind DOX and TA via hydrophobic interaction is induced. It is also found in confocal data (figure 7), showing that DOX is only incorporated in the film aggregates. Therefore, we believed that uneven CHI/TA multilayer film structure resulted in lower DOX incorporation capacity.

Even though the amount of incorporated DOX is low for CHI/TA film, sustained DOX release kinetic was confirmed. It is indirectly reflected that the DOX still bind with TA via hydrophobic interaction in PBS condition.

For better understanding to readers, we added the information and revised confocal images on the manuscript.

[Page 7, Section 2.4, line 236]

Rapid release of the DOX loaded in the CHI/HA film was observed in the release profile (Figure S4†), however, that loaded CHI/TA film was a small amount, and sustained release profile was shown. For CHI/TA film, the main interaction between TA and DOX is hydrophobic interaction (ref. Nanoscale, (2018) 10, 18228-18237.), which is hypothesized that CHI/TA film induced higher DOX incorporation efficiency as well as trapping efficiency. However, highly rough structure in CHI/TA film (Figure S3A†) that is generated by bulk CHI+TA aggregates resulted in lower chance to bind with DOX. The merged confocal image of the DOX in CHI/TA film is demonstrated that the DOX is only located in aggregates (Figure 7B). Also sustained DOX release indirectly revealed that the DOX still bind with TA via hydrophobic interaction in PBS condition (pH 7.4) which is different pH and ionic strength from film deposition condition (Figure S4†).

[Revised figure 7]

Figure 7. (A) Accumulated amount of released DOX from multilayer film under 1X PBS at 37°. (B) Confocal microscope image of DOX-incorporated multilayer films. Red fluorescence and bright field represent DOX and film structure. To be clearly seen the DOX fluorescence in film, we have adjusted red fluorescence balance. The original images are displayed in figure S6†.

Reviewer 2 Report

In this manuscript the authors provide proofs supporting their conclusions regarding the possibility of fabricating tailored-multilayer film for drug delivery application. Particularly, they found that the CHI-based multilayer films are good drug delivery system controlling the drug loading and release of three model drugs. Moreover, the authors state that these films could enhance the anticancer effect of DOX.

In my opinion this manuscript is interesting but some revisions and elucidations need to be provided.

  • Line 137: explain the meaning of “1 x” next to PBS
  • Figure 4 A: a Table should be added reporting the thickness values almost at 0 h, 30 min and 48h.
  • Figure 5: in the spectra few characteristic wavenumbers are reported. I suggest to add more characteristic values (expecially those commented in the text), moreover I suggest to indicate in each spectra the main bonds involved.
  • Line 192: Check the numeration of the supplementary figures because Figure S3 are SEM images. Figure S4: the authors should explain which kind of normalization has been performed to obtain the normalized profiles reported in this figure.
  • Figure 6A, 6B: I suggest to express on Y axis the amount in µg/cm2, as in Figure 7A.
  • Line 295: the Authors must introduce a reference regarding LbL assembly technique
  • Figure S3: it is not clear the role of this SEM analysis. The authors should clarify. Moreover, SEM analysis should be commented or cited in the manuscript otherwise consider to remove it from the supplementary figures.
  • Film Characterization by Fourier-Transform Infrared Spectroscopy is missing in the method section and it must be added in the paragraph 3.3.
  • The results of drug loading into the multilayer films obtained employing confocal laser scanning microscope are incomplete: only the images of DOX-incorporated multilayer films are shown. The authors should show the confocal images of the other two drugs.
  • I suggest, if possible, to calculate the drug entrapment efficiency for each of the drug-loaded film.

Author Response

Response to Reviewer 2 Comments

In this manuscript the authors provide proofs supporting their conclusions regarding the possibility of fabricating tailored-multilayer film for drug delivery application. Particularly, they found that the CHI-based multilayer films are good drug delivery system controlling the drug loading and release of three model drugs. Moreover, the authors state that these films could enhance the anticancer effect of DOX. In my opinion this manuscript is interesting but some revisions and elucidations need to be provided.

We would like to thank the reviewer for thorough reading of the manuscript and helpful comments. The following is point-by-point responses to the reviewer’s comments.

Point 1: Line 137: explain the meaning of “1 x” next to PBS

Response 1: Thank you for the comment. As reviewer mentioned, we added the PBS solution information at amended version.

[Page 4, Section 2.2, line 140]

The degraded thickness of the multilayer film on the Si wafer in 1× PBS (pH 7.4, 1.5 mM of KH2PO4, 0.15 M of NaCl, 2.7mM of Na2HPO4-7H2O) at 37 °C was measured using the profilometer (Figure 4A).

Point 2: Figure 4 A: a Table should be added reporting the thickness values almost at 0 h, 30 min and 48h.

Response 2: Thank you for the thoughtful comment. As reviewer pointed out, we carefully added the thickness measurement information at supporting information with independent graph.

[Revised figure S7]

Figure S7†. Table for degradation rate of multilayer films under 1× PBS at 37°C measured by changes in thickness

Point 3: Figure 5: in the spectra few characteristic wavenumbers are reported. I suggest to add more characteristic values (expecially those commented in the text), moreover I suggest to indicate in each spectra the main bonds involved.

Response 3: Thank you for the thoughtful suggestion. As reviewer mentioned, we added more FTIR characteristic wavenumbers of each multilayer film at figure 5.

[Revised figure 5]

Figure 5. FTIR spectra of (A) (CHI/HA)­­5, (B) (CHI/ALG)12, and (C) (CHI/TA)10 films.

Point 4: Line 192: Check the numeration of the supplementary figures because Figure S3 are SEM images. Figure S4: the authors should explain which kind of normalization has been performed to obtain the normalized profiles reported in this figure.

Response 4: Thank you for the thoughtful comment. As reviewer mentioned, we change the numeration of the supplementary figures S3 to S4. Also, to get normalized drug release profiles, we divide the amount of drug released at different determined time points by amount of total drug release for 34 h. We assumed that the total drug incorporation amounts are same as the accumulated for 34 hours because the release kinetics are saturated until last sample preparation time.

We added normalization information in the Section 3.4.

[Page 10, Section 3.4, line 345]

The normalized drug release profiles were calculated by division the amount of drug released at different determined time points by amount of drug release for 34 h.

Point 5: Figure 6A, 6B: I suggest to express on Y axis the amount in µg/cm2, as in Figure 7A.

Response 5: Thank you for the thoughtful suggestion. As reviewer mentioned, we expressed on Y axis the amount in µg/cm2, as in figure 6A, 6B.

[Revised figure 6]

Figure 6. Accumulated amount of released (A) FITC and (B) Ova from multilayer films under 1× PBS at 37 °C.

Point 6: Line 295: the Authors must introduce a reference regarding LbL assembly technique

Response 6: Thank you for the thoughtful suggestion. We agreed for lack of references about layer-by-layer (LbL) assembly. We added several related LbL publications at amended version of manuscript.

[Revised references, line 310]

  1. Decher, Science, 1997, 277, 1232.
  2. S. Shiratori, M. F. Rubner, Macromolecules, 2000, 33, 4213.
  3. Wang, L. B. Zhang, L. Wang, J. Q. Sun, J. C. Shen, Langmuir, 2010, 26, 8187.

Point 7: Figure S3: it is not clear the role of this SEM analysis. The authors should clarify. Moreover, SEM analysis should be commented or cited in the manuscript otherwise consider to remove it from the supplementary figures.

Response 7: Thank you for the comment. In case of CHI/TA film, the film grows highly rough and uneven, which is clearly affected their thickness growth curve. Therefore, we tried to measure the morphology of CHI/TA film in a large area. SEM images can cover the macroscopic morphology rather than AFM images (5×5 mm), we could confirm that the film has generated by CHI and TA aggregates in bulk, not evenly distributed film in a molecular level. These film structure further affects DOX incorporation, the DOX embedded only in aggregates (figure 7).

Also, after film degradation (figure S3B), the degradation of CHI+TA aggregates are observed. dis-linking and disassembling aggregates are clearly shown as indicated in white arrows.

According to the reviewer’s comment, we added the sentences to back up the morphology, disassembly and DOX incorporation capacity of CHI/TA film using SEM images.

[Page 4, Section 2.1, line 129]

High surface roughness was observed in the CHI/TA film because the rarely ionized CHI chain enabled the CHI/TA multilayer to form a rough and macroporous structure (Figure S3†). The macroscopic images of CHI/TA film revealed that macro-size clumps are formed, whereas non- or little-coated areas also existed.

[Page 5, Section 2.2, line 157]

The dis-linking and disassembling of CHI+TA aggregates which are indicated in Figure S3B† (white arrows) are clearly existed, demonstrating that the CHI/TA film are degraded in macroscopic range.

[Page 7, Section 2.4, line 240]

However, highly rough structure in CHI/TA film (Figure S3A†) that is generated by bulk CHI+TA aggregates resulted in lower chance to bind with DOX.

Point 8: Film Characterization by Fourier-Transform Infrared Spectroscopy is missing in the method section and it must be added in the paragraph 3.3.

Response 8: Thank you for the thoughtful comment. As reviewer mentioned, we added the method of Fourier-Transform Infrared Spectroscopy in the Section 3.3.

[Pahe 10, Section 3.3, line 323]

The film characteristics for functional groups were investigated by Fourier-Transform Infrared Spectroscopy (IR-4700, Jasco, Japan) using dried CHI-base films. The infrared spectra of film characteristics for functional group was recorded on a Jasco spectrometer by averaging 32 scans with a resolution of 4 cm-1.

Point 9: The results of drug loading into the multilayer films obtained employing confocal laser scanning microscope are incomplete: only the images of DOX-incorporated multilayer films are shown. The authors should show the confocal images of the other two drugs.

Response 9: Thank you for the thoughtful comment. In this study, we have measured the confocal image only for DOX. According to the confocal images, we could find out the deposited location of DOX onto the film and relative incorporation amount of DOX under same laser intensity, not quantitative amount. Therefore, we additionally confirmed release kinetics of 3 kinds of drugs to quantify. Also, the DOX deposition within nanosized multilayer film could be observed roughly by confocal microscope which is microscale resolution. For DOX incorporated film, the DOX locates only onto the multilayer films and aggregates. Therefore, this information could be backed up by other data, release kinetics and SEM or AFM data. For example, the CHI/TA film has highly aggregated structure, the incorporated FITC and Ova could be displayed only in the film aggregates in moderate amounts (see FITC and Ova release kinetics).

Point 10: I suggest, if possible, to calculate the drug entrapment efficiency for each of the drug-loaded film.

Response 10: Thank you for the comment. In terms of entrapment efficiency, we treated extremely concentrated drug solution to safely saturate incorporation within 1 hour, therefore, the ratio treated drug to incorporated drug (entrapment efficiency) is quite low. For example, DOX entrapment efficiency for CHI/ALG film is around 1.12% because we used 0.5 mg/mL of DOX solution for treatment. However, we believed that we could control the highly cell-sensitive drug (i.e. DOX) incorporation amount by changing counter polymers (figure 8), which represents that we successfully prepared efficient CHI-based multilayer film as a drug reservoir.

Round 2

Reviewer 2 Report

The manuscript can be now accepted in the present form